# Co-Infections in Critically Ill Patients with or without COVID-19: A Comparison of Clinical Microbial Culture Findings

**DOI:** 10.3390/ijerph18084358

**Published:** 2021-04-20

**Authors:** Rosario Cultrera, Agostino Barozzi, Marco Libanore, Elisabetta Marangoni, Roberto Pora, Brunella Quarta, Savino Spadaro, Riccardo Ragazzi, Anna Marra, Daniela Segala, Carlo Alberto Volta

**Affiliations:** 1Infectious Diseases, Department of Traslational Medicine, University of Ferrara, 44121 Ferrara, Italy; daniela.segala@unife.it; 2Clinical Microbiology, Department of Biotechnology, Transfusional and Laboratory, University Hospital of Ferrara, 44124 Ferrara, Italy; a.barozzi@ospfe.it (A.B.); roberto.pora@ospfe.it (R.P.); 3Infectious Diseases Unit, Department of Medicine, University Hospital of Ferrara, 44124 Ferrara, Italy; m.libanore@ospfe.it; 4Intensive Care Unit, Department of Traslational Medicine, University of Ferrara, 44121 Ferrara, Italy; e.marangoni@ospfe.it (E.M.); spdsvn@unife.it (S.S.); rgc@unife.it (R.R.); vlc@unife.it (C.A.V.); 5Pharmacy Service, University Hospital of Ferrara, 44124 Ferrara, Italy; b.quarta@ospfe.it (B.Q.); a.marra@ospfe.it (A.M.)

**Keywords:** SARS-CoV-2, COVID-19, co-infection, microbial culture, antimicrobial consumption, antimicrobial expenditure, intensive care unit

## Abstract

Co-infections in critically ill patients hospitalized for severe acute respiratory syndrome coronavirus 2 (SARS-CoV-2) infection have an important impact on the outcome of coronavirus disease 2019 (COVID-19). We compared the microbial isolations found in COVID-19 patients hospitalized in an intensive care unit (ICU) with those in a non-COVID-19 ICU from 22 February to 30 April 2020 and in the same period of 2019. We considered blood, urine or respiratory specimens obtained with bronchoalveolar lavage (BAL) or bronchial aspirate (BASP), collected from all patients admitted in ICUs with or without COVID-19 infection. We found a higher frequency of infections due to methicillin-resistant (MR) staphylococci, vancomycin-resistant *Enterococcus faecium*, carbapenem-resistant *Acinetobacter baumannii* and *Candida parapsilosis* in COVID-19-positive patients admitted in ICUs compared to those who were COVID-19 negative. Carbapenem-resistant *Pseudomonas aeruginosa* was more frequently isolated from patients admitted in non-COVID-19 ICUs. Several conditions favor the increased frequency of these infections by antibiotic-resistant microorganisms. Among all, the severity of the respiratory tracts was definitely decisive, which required assisted ventilation with invasive procedures. The turnover in the ICU of a large number of patients in a very short time requiring urgent invasive interventions has favored the not always suitable execution of assistance procedures. No less important is the increased exposure to infectious risk from bacteria and fungi in patients with severe impairment due to ventilation. The highest costs for antifungal drugs were shown in the ICU-COVID group.

## 1. Introduction

Patients admitted to intensive care units (ICUs) are exposed to a greater risk of secondary infectious complications that more frequently affect the respiratory, urinary and circulatory systems, constituting one of the main causes of healthcare-associated infections (HAIs). These infections are often due to multi-drug-resistant microorganisms that require the use of next-generation antimicrobials with a serious impact on antimicrobial stewardship programs, on the appropriateness of the use of antibiotics and on a pharmaco-economic level. The microbial analyses performed on these patients also contribute to the increase in healthcare costs, and are often performed in large quantities and not always necessary, resulting in their misinterpretation. In fact, the complicated distinction between colonization, contamination of the biological sample taken and infection often causes an overestimation of infections with consequent inappropriate antibiotic therapy. Bacterial and fungal co-infections can be complications of viral respiratory diseases such as seasonal/pandemic influenza [1], Middle East respiratory syndrome coronavirus (MERS-CoV) [2] and SARS-CoV-1 [3,4], and lead to an increase in outcome severity and to the death of individuals with or without pre-existing respiratory diseases.

The clinical manifestations of coronavirus disease 2019 (COVID-19) range from asymptomatic infection to severe viral pneumonia requiring treatment in an intensive care unit (ICU) [5,6,7]. The most common symptoms of COVID-19 are fever and cough and, due to serious respiratory disease, some patients need to be hospitalized and mechanical ventilation [6]; more severe outcomes [8] are associated with older age, a higher percentage of comorbidities and higher mortality [9,10]. 

SARS-CoV-2 can directly damage the lung epithelium and indirectly ignite an aberrant cytokine storm, eventually leading to multi-organ failure [11,12]. To reverse this dysregulated activation of the immune system, immunosuppressive drugs are widely used [12,13]. A combination of virus- and drug-induced immunosuppression likely increases the susceptibility to secondary infections. 

Some patients with COVID-19 need to be hospitalized due to severe respiratory complications and, in severe cases, undergo intensive care with supportive mechanical ventilation. Bacterial and fungal co-infections in critically ill patients hospitalized for SARS-CoV-2 infection have an important impact on the outcome of COVID-19 disease, with increased morbidity and mortality [14,15,16].

There are numerous studies on the empirical use of antibiotics in hospitalized patients with COVID-19 [17,18], often induced by the evidence of inflammatory markers such as procalcitonin and C-reactive protein, which are normally associated with bacterial infections, even when they are not proven [19,20]. 

Our study examined microbial cultures performed in COVID-19-positive patients admitted to an ICU from 22 February 2020 to 30 April 2020. The results were compared with microbial cultures performed in hospitalized non-COVID-19 patients in another ICU of the same hospital from 22 February 2020 to 30 April 2020 and in 2019, prior to the pandemic spread of COVID-19. We also analyzed the consumption of antibiotics in these ICUs in the same periods.

The primary objectives were to define (i) the number of microbial cultures of blood, urine and bronchoaspirate/bronchoalveolar lavage samples, (ii) the frequency of isolated bacterial and fungal species, (iii) the consumption of antimicrobials in ICUs in the two periods examined and in patients with or without COVID-19.

The secondary objective was to evaluate the expenditure on antimicrobials used by each ICU in relation to the infections identified by culture tests.

## 2. Materials and Methods

### 2.1. Study Population

We considered blood, urine or respiratory specimens obtained with bronchoalveolar lavage (BAL) or, when it was not available, bronchial aspirate (BASP), collected from all patients admitted in intensive care units (ICUs) with (ICU-COVID) or without COVID-19 infection (ICU-noCOVID) from 22 February to 30 April 2020 to study the microbiological findings. Moreover, we considered the same specimens collected in the ICU (ICU-2019) from 22 February to 30 April 2019, before the pandemic. 

The ICUs were mixed medical–surgical ICUs (ICU-noCOVID and ICU-2019), and a dedicated ICU for COVID-19-positive patients (ICU-COVID) at University Hospital of Ferrara, Italy. All patients admitted in these ICU wards needed mechanical ventilation settings, including constant-flow controlled ventilation.

COVID-19 was defined as the presence of a positive real-time polymerase chain reaction (RT-PCR) for SARS-CoV-2 in at least one respiratory specimen (nasopharyngeal swab, sputum, and/or lower respiratory tract specimens), associated with suggestive signs, symptoms and/or radiological findings. 

Bloodstream infection (BSI) was defined as the presence of at least one positive blood culture for bacteria or fungi, drawn >48 h after ICU admission. At least two consecutive positive blood cultures were needed to define BSI due to coagulase-negative staphylococci or other common skin colonizers (e.g., coagulase-negative staphylococci, non-diphtheritic *Corynebacteria*, *Bacillus* spp., *Propionibacterium* spp., etc.) [21]. Patients who had more than one positive blood culture within 7 days from the first positive blood culture were considered to have a single episode of BSI with multiple isolates. Polymicrobial infections were considered as separate BSI events, one for each causative organism isolated from blood culture. 

Positive cultures of potentially pathogenic organisms from the lower respiratory tract were defined as positive cultures of a respiratory specimen obtained with BAL or, when not available, BASP, excluding *Candida* spp. 

Microbiologically, catheter-associated urinary tract infection (CA-UTI) is defined by microbial growth of ≥10^3^ colony-forming units (CFU)/mL of one or more bacterial species in a single catheter urine specimen or in a midstream voided urine specimen from a patient whose urethral, suprapubic or condom catheter has been removed within the previous 48 h. All criteria to confirm CA-UTI in several clinical conditions and with particular microbiological findings were considered according to international classifications and reports [22,23,24]. 

Infection versus colonization was distinguished according to definitions reported by the European Centre for Disease Prevention and Control [25,26].

Each isolate was identified by matrix-assisted laser desorption ionization time-of-flight (MALDI-TOF) by a VITEK^®^ MS (bioMerieux). Antibiotic susceptibility testing was performed by a VITEK^®^ 2 instrument (bioMerieux), with Card AST-N376 for Gram-negative bacteria screening and N397 to test the susceptibility of *Enterobacteriales* to cephalosporins/β-lactamase inhibitor combinations, Card AST-P659 for staphylococci, Card AST-P658 for enterococci, Card AST-ST03 for streptococci (*S. pneumoniae*, beta-hemolytic *Streptococcus* and viridans *Streptococcus*) and Card AST-YS08 for clinically significant yeasts.

Methicillin-resistant *Staphylococcus aureus* (MRSA) and vancomycin-resistant Enterococci (VRE) strains were detected by cefoxitin screening and an oxacillin MIC test with Card AST-P659 and a teicoplanin/vancomicin MIC test with Card AST-P658, respectively.

MIC values were interpreted according to current European Committee on Antimicrobial Susceptibility Testing (EUCAST) clinical breakpoints.

Carbapenemase-producing *Enterobacteriales* (CPE) strains were confirmed by microdilution (Sensititre™, Thermo Fisher Scientific) EURGNCOL and DKMGN plates. Phenotypical CPE resistance was confirmed by synergic test diffusion Diatabs™ (Rosco Diagnostica) on Mueller–Hinton agar (Vacutest-Kima). A genotyping test for CPE resistance was performed by RT-PCR (GeneXpert^®^) with an Xpert^®^ Carba-R test (Cepheid Inc., Sunnyvale, CA, USA). MIC values were interpreted according to current European Committee on Antimicrobial Susceptibility Testing (EUCAST) clinical breakpoints.

Blood, respiratory and urine cultures were requested by the attending physicians for patients with suspected secondary infections because of clinical and/or respiratory deterioration associated with suggestive laboratory or radiological findings.

### 2.2. Antimicrobial Consumption and Costs

The data of the antibiotic consumption were normalized to DDD/100 bed days for inpatients. This information was calculated independently by the pharmacy service, based on dispensing rather than administration data, using our own computer system that estimated the DDD [27].

Crude costs of antimicrobial drugs were collected from the administrative pharmacy data.

### 2.3. Statistical Analyses 

We described the microbial culture carried out in ICU wards with or without COVID-19-positive patients between 6 March and 30 April 2020 compared to the microbial tests done in same ward during the same period in 2019. Descriptive analyses were performed for all variables. Count and proportion were presented for all categorical variables. 

The primary outcomes were the number of microbial cultures and the frequency of bacterial and fungal species isolated in ICUs with or without COVID-19 positive patients, and to define the consumption of antimicrobials in these wards.

The secondary aim was to evaluate a possible higher expenditure on antimicrobials in ICUs during the pandemic.

Bivariate comparisons using chi-square (or Fisher’s exact) tests were conducted for nominal data and two-sample *t*-tests or Mann–Whitney U tests for continuous data (depending on normality distribution) were used to compare characteristics and outcomes between the samples of patients. A *p* value < 0.05 was accepted as statistically significant.

## 3. Results

The analyses performed show an increase in the number of microbial cultures per patient done in the ICU-COVID (20.5/patient) group compared to the number of analyses made in the same period in the ICU-noCOVID group (7/patient) and in ICU-2019 group (8.3/patient), before the SARS-CoV-2 pandemic (Table 1). The clear difference remains even after subtracting from the count the double negative samples on the same day (ICU-COVID: 14.5/patient; ICU-noCOVID: 4.4/patient; ICU-2019: 5.5/patient) (Table 2). There was an increase in positive blood cultures compared to the other two cohorts, while positive culture differences were not found for bronchoaspirate, broncholavage and urine cultures (Table 1).

Taking into consideration the different classes of isolated microorganisms, an increase in fungal infections in ICU-COVID emerges compared to the other two cohorts: *Candida albicans* (*n* = 29), *C. parapsilosis* (*n* = 13). The most frequently isolated bacteria were *Acinetobacter baumannii*, *Enterococcus faecalis* and *E. faecium*, *Staphylococcus epidermidis* and *Stenotrophomonas maltophilia* (Table 3). No relevant data emerged regarding the number of positive cultures for MRSA, extended spectrum β-lactamase (ESBL)-positive and carbapenem-resistant and carbapenemase-producing *Enterobacteriaceae* in the three cohorts (Table 4).

Consumption of antimicrobials significantly increased in the ICU-COVID group compared to the other two cohorts. In parallel with these data, there is an increase in DDD/100 bed days for meropenem, piperacillin/tazobactam, colistin, vancomycin and caspofungin in the ICU-COVID group (Table 5).

Costs for antimicrobials increased in ICU-COVID, with particular reference to carbapenems, piperacillin/tazobactam, colistin, linezolid, tigecycline and third-generation cephalosporins, while there was no increase in ceftazidime/avibactam compared to the previous year. There was an increased use of liposomal amphotericin B in patients with COVID-19 compared to that recorded in the same period in patients without COVID-19 and in those hospitalized in 2019. The ICU-COVID expenditure on antimicrobials amounted to EUR 58,264, an increase of 341% and 605.1% compared to costs in the same period of 2019 and 2020 in the ICU-2019 group and in the ICU-noCOVID group, respectively (Table 6).

The highest costs were found in the ICU-COVID ward for liposomal amphotericin B and echinocandins.

The increase in costs was strictly related to the increased consumption of antimicrobials and not to the increase in direct costs or the reduction in drug supply affected by the lockdown.

## 4. Discussion

We studied the microbial culture findings to evaluate the frequency of isolated bacteria and fungi in different specimens obtained from patients hospitalized in ICUs with or without COVID-19-positive patients between 6 March and 30 April 2020 and in the same ICU from March to April 2019. Mortality due to secondary infections in COVID-19 patients in Wuhan was reported to be 16% hospitalized patients [28]. 

The incidence rate of BSIs appears significantly higher compared to those concerning the other two cohorts. BSIs constituted the majority of secondary infections, with 71/206 positive cultures from the central venous blood system and 48/198 positive cultures from the peripheral venous blood system (*p* < 0.05). We documented a higher rate of Gram-positive isolates in blood cultures. These findings may reflect a high burden of catheter-associated blood infections, in particular those due to *Staphylococcus epidermidis*.

No difference of positive cultures in respiratory and urinary specimens in three cohorts was shown. 

Lower respiratory tract infections were documented for 88/145 samples of respiratory tract cultures, mainly due to *Candida* spp. and Gram-negative bacteria. The identification of Gram-negative organisms is consistent with the types of pathogens frequently associated with hospital-acquired pneumonia (HAP) in ICUs as a complication of ICU care in both pre-COVID-19 periods [29] and during the pandemic [30,31,32,33]. 

The large reduction of cultures from the lower respiratory tract compared to microbial isolates from the upper respiratory tract allows us to hypothesize that yeast and bacterium-positive respiratory tract cultures were mostly likely due to contamination/colonization of the respiratory devices other than possible commensal microorganisms of the upper respiratory tract.

These data could be related to multiple device management in a pandemic setting which may have reduced the adherence to strict aseptic procedures, especially in critically ill patients managed outside the ICU or in an overcrowded ICU. Furthermore, the proper use of personal protective equipment can be challenging and can lead to lower compliance with aseptic techniques in the management of intravascular devices, which was also favored by the recruitment of new emergency health personnel. Our results are consistent with other reports from similar cohorts where a higher frequency of BSIs was reported but differ from those of a higher percentage of Gram-negative in positive respiratory cultures revealed in our study [34].

Overall, the most frequently isolated microorganisms in an ICU-COVID ward compared to the other two cohorts in our study were *Acinetobacter baumannii, Enterococcus faecium, E. faecalis, S. epidermidis, Stenotrophomonas maltophilia, Candida albicans* and *C. tropicalis*. Our data showed that increased numbers of microbial analyses and positive cultures were matched by an increased consumption of antimicrobial drugs, calculated as expenditure per drug and as DDD/100 bed days. The overall expenditure on antimicrobials between March and April 2020 in the ICU-COVID ward was EUR 58,264 compared to EUR 9629 and EUR 17,086 spent in the ICU-noCOVID and ICU-2019 wards in 2020 and 2019, respectively.

The most used drugs were piperacillin/tazobactam (44.7 DDD/bed days), meropenem (31.9 DDD/bed days), caspofungin (22.5 DDD/bed days), vancomycin (17.1 DDD/bed days) and colistin (5.9 DDD/bed days). 

The finding of an expenditure for liposomal amphotericin B in the COVID-ICU ward that is not comparable with the other two cohorts confirms the data of a higher frequency of fungal isolations in patients with COVID-19, in agreement with what was reported by other authors [35,36,37].

These data show that in the course of the SARS-CoV-2 pandemic, there was a greater consumption of antibiotics but also an inappropriate dosage, calculated as DDD/100 days of hospitalization, in comparison with what was demonstrated in the ICU-2019 ward in 2019 and in the ICU-noCOVID ward in 2020. Other reports did not support the routine use of antibiotics in the management of COVID-19 co-infections because of little evidence of bacterial co-infections [38]. Our experience suggests that antimicrobial stewardship strategies should be targeted to limiting unnecessary antimicrobial employment during the pandemic [39].

Our study has some limitations. First of all, the limited period examined did not allow a numerically larger sample to be examined. The study was primarily aimed at defining the number of microbial tests required by an intensive care unit during the SARS-CoV-2 pandemic. These data were then compared with the microbial analyses carried out by the same SARS-CoV-2-negative ICU in the same periods of 2020 and 2019. Second, data regarding empirical antimicrobial use were not available, limiting the assessment of its impact on the development of secondary infections. Third, it was impossible to distinguish the persistence of positivity of the microbial analyses from the incorrect frequent repetition of the analyses for each patient, which can lead to an overestimation of infections.

## 5. Conclusions

Co-infections are possible in COVID-19-positive patients hospitalized in ICUs. Our study is based on a limited number of microbiological data collected in a short period. The increase in culture analyses performed in positive COVID-19 patients admitted in an ICU ward compared to patients without SARS-CoV-2 infection leads to the hypothesis of an excessive use of microbial analyses with numerous and frequent sampling. It remains of major importance to distinguish colonization from infection. Their misunderstanding is a major cause of inappropriate antimicrobial therapy in the absence of further clinical data that can confirm bacterial or yeast co-infection.

Our study shows how important it is to have an antimicrobial stewardship program which is useful in pandemic emergencies, such as the current one, in order to contribute to a better use of economic resources particularly in regard to those for laboratory analyses and drugs.

The need to implement responsible antimicrobial stewardship is confirmed in order to limit unnecessary use of antimicrobials and limit the trend of antibiotic resistance.

## Figures and Tables

**Table 1 ijerph-18-04358-t001:** Microbial cultures in patients.

	ICU-2019		ICU-noCOVID		ICU-COVID		*p*
	Patients (*n* = 58)		Patients (*n* = 47)		Patients (*n* = 28)		
	TOT	POS (%)	TOT	POS (%)	TOT	POS (%)	
Central venous blood culture	178	26 (14.6)	119	18 (15.1)	206	71 (34.5)	<0.05
Periferal venous blood culture	160	21 (13.1)	122	12 (9.8)	198	48 (24.3)	<0.05
Bronchoaspirate culture	103	52 (50.5)	72	40 (55.6)	134	82 (61.2)	N.S.
Bronchoalveolar lavage culture	15	7 (46.7)	8	4 (50)	9	6 (66.7)	N.S.
Urinary catheter	25	10 (40)	7	4 (57.1)	27	13 (48.1)	N.S.
Tot.	481	116 (24.1)	328	78 (23.8)	574	220 (38.3)	<0.05

N.S.: not significant, *p* > 0.05.

**Table 2 ijerph-18-04358-t002:** Gram-positive and Gram-negative bacteria and fungal isolates.

	ICU-2019			ICU-noCOVID			ICU-COVID		
	Patients (*n* = 58)			Patients (*n* = 47)			Patients (*n* = 28)		
	Gram+	Gram−	Fungi	Gram+	Gram−	Fungi	Gram+	Gram−	Fungi
Central venous blood culture	21	4	1	13	5	0	43	16	12
Periferal venous blood culture	16	5	0	10	2	0	32	10	6
Bronchoaspirate culture	3	46	3	3	17	20	9	50	23
Bronchoalveolar lavage culture	2	5	0	1	1	2	3	3	0
Urinary catheter	2	7	1	0	1	3	2	2	9
Tot.	44	67	5	27	26	25	89	81	50

**Table 3 ijerph-18-04358-t003:** Bacterial and fungal isolates in patients admitted in three ICUs.

	ICU-2019	ICU-noCOVID	ICU-COVID	*p*
	Patients (*n* = 58)	Patients (*n* = 47)	Patients (*n* = 28)	
Microbial Isolates	*n*	*n*	*n*	
*Acinetobacter baumannii*	9	-	17	<0.05
*Candida albicans*	-	11	29	<0.05
*Candida glabrata*	2	5	4	N.S.
*Candida krusei*	-	1	-	N.S.
*Candida lusitaniae/Ciavispora I*	2	-	2	N.S.
*Candida parapsilosis*	-	-	13	N.S.
*Candida tropicalis*	1	8	-	N.S.
*Enterococcus faecalis*	2	1	14	<0.05
*Enterococcus faecium*	1	1	10	<0.05
*Pseudomonas aeruginosa*	15	9	3	N.S.
*Staphylococcus epidermidis*	21	10	42	<0.05
*Stenotrophomonas maltophilia*	3	5	13	<0.05
Total Samples Pos.	116	78	120	<0.05

N.S.: not significant, *p* > 0.05.

**Table 4 ijerph-18-04358-t004:** Resistant bacteria isolated in three cohorts.

	ICU-2019			ICU-noCOVID			ICU-COVID		
	Patients (*n* = 58)			Patients (*n* = 47)			Patients (*n* = 28)		
	MRSA	CPE/CRE	ESBL+	MRSA	CPE/CRE	ESBL+	MRSA	CPE/CRE	ESBL+
Central venous blood culture	1	0	1	0	0	0	0	1/1	0
Periferal venous blood culture	0	0	1	0	0	0	0	0	0
Bronchoaspirate culture	2	0	0	2	1/1	0	2	2/2	0
Bronchoalveolar lavage culture	1	0	0	0	0	0	3	0	0
Urinary catheter	0	0	0	0	0	0	0	1/1	0
Tot.	4	0	2	2	1/1	0	5	4/4	0

MRSA, methicillin-resistant *Staphylococcus aureus*; CPE, carbapenemase-producing *Enterobacteriales*; CRE, carbapenem-resistant *Enterobacteriales*; ESBL, extended spectrum β-lactamase.

**Table 5 ijerph-18-04358-t005:** DDD/100 bed days.

	ICU-2019	ICU-noCOVID	ICU-COVID
	March–April 2019	March–April 2020	March–April 2020
	Patients (*n* = 58)	Patients (*n* = 47)	Patients (*n* = 28)
Amoxicillin/clavulanic acid	84.5	105.8	6.5
Azithromycin	0	14.8	10.9
Ceftriaxone	9.5	18.1	2
Colistin	0	0	5.9
Daptomycin	3.4	20	0
Meropenem	7.9	23	31.9
Piperacillin/tazobactam	14.3	37.6	44.7
Vancomycin	0	11.5	17.1
Caspofungin	10	8.2	22.5
Fluconazole	1.7	32.9	0
Voriconazole	-	-	-

DDD: defined daily dose.

**Table 6 ijerph-18-04358-t006:** Costs of specific antibiotic drugs in three ICU cohorts.

	ICU-2019	ICU-noCOVID	ICU-COVID
	Patients (*n* = 58)	Patients (*n* = 47)	Patients (*n* = 28)
Aminoglycosides	EUR 51	EUR 45	EUR 17
Amoxicillin/clavulanic acid	EUR 862	EUR 690	EUR 87
Carbapenems	EUR 259	EUR 544	EUR 1268
Ceftazidime/avibactam	EUR 3256	EUR 0	EUR 2442
Ceftolozane/tazobactam	EUR 774	EUR 0	EUR 0
Cephalosporins I-II	EUR 0	EUR 67	EUR 0
Cephalosporins III	EUR 248	EUR 283	EUR 616
Fluoroquinolones	EUR 51	EUR 46	EUR 0
Glycylcyclines	EUR 752	EUR 280	EUR 840
Glycopeptides	EUR 0	EUR 134	EUR 335
Lincosamides	EUR 64	EUR 96	EUR 0
Lipopeptides	EUR 629	EUR 709	EUR 0
Macrolides	EUR 233	EUR 361	EUR 276
Nitroimidazoles	EUR 54	EUR 54	EUR 0
Linezolid	EUR 231	EUR 144	EUR 867
Penicillin	EUR 15	EUR 0	EUR 0
Piperacillin/tazobactam	EUR 1020	EUR 1079	EUR 3259
Colistin	EUR 0	EUR 0	EUR 210
Sulfonamides	EUR 0	EUR 4	EUR 0
Liposomal amphotericin B	EUR 0	EUR 0	EUR 24,956
Triazoles	EUR 1	EUR 47	EUR 0
Echinocandins	EUR 8586	EUR 5046	EUR 23,091
Tot.	EUR 17,086	EUR 9629	EUR 58,264

## Data Availability

The data presented in this study are contained within the article and they are available on a reasoned request by contacting the reference author.

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
