# Peer review of "Co-Infections in Critically Ill Patients with or without COVID-19: A Comparison of Clinical Microbial Culture Findings"

_ijerph, 2021, doi:10.3390/ijerph18084358_

Round 1

Reviewer 1 Report

Major

1. Authors studied about this topic at Intensive Care Units (ICUs). 

Could you inform what kind of ICU to study in manuscript? 

Because several kinds of ICU are operating and guess that the infection source may be different according to kind of ICU.  

2. Between line 211 and 215, author describe below 

“The large reduction of cultures from the lower respiratory tract compared to microbiological isolates from the upper respiratory tract allows us to hypothesize that yeast and bacterial positive respiratory tract cultures were mostly likely due to contamination/colonization of the respiratory devices”

I agree with the authors and I also think that CoVID 19 patients also had opportunistic infection by commensal microorganism in the upper respiratory tract.

Do you have some data about opportunistic infection by commensal microorganism in the respiratory tract?   

Minor

1. In line 17, 18, and 21, please write the full name of SARS-CoV-2, CoVID-19, and MR.

2. In line 49, please check “)”

Author Response

Reviewer 1:

Major

  1. Authors studied about this topic at Intensive Care Units (ICUs). 

Could you inform what kind of ICU to study in manuscript? 

Because several kinds of ICU are operating and guess that the infection source may be different according to kind of ICU.  

1R. The ICUs were a mixed medical-surgical ICU and a dedicated COVID-19 ICU at University Hospital of Ferrara, Italy. All their patients needed mechanical ventilation settings included constant-flow controlled ventilation.

  1. Between line 211 and 215, author describe below “The large reduction of cultures from the lower respiratory tract compared to microbiological isolates from the upper respiratory tract allows us to hypothesize that yeast and bacterial positive respiratory tract cultures were mostly likely due to contamination/colonization of the respiratory devices”.

I agree with the authors and I also think that COVID 19 patients also had opportunistic infection by commensal microorganism in the upper respiratory tract.

Do you have some data about opportunistic infection by commensal microorganism in the respiratory tract?   

2R. This sentence has been modified as “The large reduction of cultures from the lower respiratory tract compared to microbiological isolates from the upper respiratory tract allows us to hypothesize that yeast and bacterial positive respiratory tract cultures were mostly likely due to contamination/colonization of the respiratory devices other than a possible commensal microorganisms of the upper respiratory tract”.

Minor

  1. In line 17, 18, and 21, please write the full name of SARS-CoV-2, COVID-19, and MR.

1R. Full names have been added: severe acute respiratory syndrome Coronavirus-2 (SARS-CoV-2) infection; Coronavirus disease 2019 (COVID-19; methicillin resistant (MR)-Staphylococci.

  1. In line 49, please check “)”

2R. Bracket has been added. (COVID-19)

Reviewer 2 Report

The manuscript by Cultrera et al., entitled “Co-Infections in Critically Ill Patients with or without CoVID-2 19: a Comparison of Clinical Microbiological Culture Findings” reports on the rate of coinfection in critically ill patients with or without COVID-19. The authors analyzed the number of microbial cultures performed (total/positive), the taxa associated, and the rate of antibiotic drugs used. The manuscript is descriptive. The main highlights and findings are unclear.

Major comments:

-The abstract hardly summarize the methods and results shown within the manuscript. The biological matrices assessed should be mentioned. The main findings are not clearly highlighted in the abstract (nor the conclusion). E.g. authors state in the abstract: “We found a higher frequency of infections due to MR-Staphylococci, vancomycin-resistant Enterococcus faecium, carbapenem-resistant Pseudomonas aeruginosa and Acinetobacter baumannii in CoVID-19 positive patients admitted in ICU compared to those CoVID-19 negative.” However, in the results section authors mentioned that “No relevant data emerged regarding the number of positives for MRSA”; There is no mention of MRSA or VRE or carbapenem-resistant Pseudomonas in the text. The abstract (and conclusion) should be improve to clearly highlight the main findings and relevance.

-In M&M, the paragraph on CA-UTI (line 108-line124) is much too detailed, and should be shortened using a reference while mentioning the main microbiological reading used. For e.g., Line 118, Are the signs and symptoms compatible with CA-UTI crucial for the present study (or relevant in the context of critically ill COVID-19 patients?). In M&M, authors should also clarify 1) how resistance to antibiotics were performed and 2) how the drug costs (economic analysis) were calculated.

- The names of the three cohorts are very similar, and as such slightly confusing, i.e. COVID-19 ICU, no-COVID-19 ICU, and ICU-2019. The name also changes from COVID-19 ICU to ICU-COVID-19, and towards the end of the manuscript as SARS-CoV-2 ICU. Please harmonize throughout the text.

-There is no information about the patients in the three cohorts, therefore it is difficult to estimate how comparable they are. Please provide more information on patients characteristics in M&M. Any information available related to the conditions/causes for the admission of “no-COVID-19 ICU” and “ICU-2019” patients. Admission for other respiratory infections? Were these patients also connected to respiratory support system or other support systems?

-There is no information about data collection; from where does these data come from? Hospital?

-In the current form, the discussion does not provide added-value compare to the results section. It mostly paraphrase the results section. Only 2 references are used throughout the full discussion.

-Table 4 is not mentioned in the text. Please adapt.

- The first sentence of the conclusion is rather surprising, as authors mentioned that respiratory co-infection cannot be excluded in COVID-19 patients, but their data show no significant difference between the three cohorts. The structure should be adapted, highlighting the main findings.

Specific comments:

Line 49: “The clinical manifestations of Coronavirus disease 2019, COVID-19)” => “The clinical manifestations of Coronavirus disease 2019 (COVID-19)”

Line 18, Line 49, and throughout text: It is not clear why authors use both CoVID-19 and COVID-19 nomenclature (when referring to the disease). Homogenized throughout text using WHO nomenclature for both disease and virus: https://www.who.int/emergencies/diseases/novel-coronavirus-2019/technical-guidance/naming-the-coronavirus-disease-(covid-2019)-and-the-virus-that-causes-it

Line 71: “patients. In" =>  "patients in"

Line 75: “Primary objectives were (i) the number of microbiological […]” => Please adapt this sentence containing the objectives. A word is likely missing.

Line 113: Define CA-ABU

Line 115: "CA-asymptomatic bacteriuria" => "CA-asymptomatic bacteriuria (CA-ASB)"

Line 149: “in the 2019” => “in 2019”

Line 160: Please clarify the following sentence: “No relevant data emerged regarding the number of positives for MRSA, ESBL-positive and carbapenem-resistant and carbapenemase-producing Enterobacteriaceae in the three cohorts." No colonies were isolated or no difference of CFU/mL between the cohorts?

Line 164: Define DDD in text here (or prior in M&M).

Line 166: The tables numbering is odd; Going from table 3 directly to table 6 in the text. Please adapt with incrementing numbers.

Line 219:  […] may have to led to reduced compliance with […] => Clumsy English. Please adapt.

Line 234: In this list, order the drug in a logical manner (e.g. from most used to least used).

Table 6: “Pazients” => “Patients”

Author Response

Reviewer 2:

The manuscript by Cultrera et al., entitled “Co-Infections in Critically Ill Patients with or without COVID-2 19: a Comparison of Clinical Microbiological Culture Findings” reports on the rate of coinfection in critically ill patients with or without COVID-19. The authors analyzed the number of microbial cultures performed (total/positive), the taxa associated, and the rate of antibiotic drugs used. The manuscript is descriptive. The main highlights and findings are unclear.

Major comments:

1.The abstract hardly summarize the methods and results shown within the manuscript. The biological matrices assessed should be mentioned. The main findings are not clearly highlighted in the abstract (nor the conclusion). E.g. authors state in the abstract: “We found a higher frequency of infections due to MR-Staphylococci, vancomycin-resistant Enterococcus faecium, carbapenem-resistant Pseudomonas aeruginosa and Acinetobacter baumannii in COVID-19 positive patients admitted in ICU compared to those COVID-19 negative.” However, in the results section authors mentioned that “No relevant data emerged regarding the number of positives for MRSA”; There is no mention of MRSA or VRE or carbapenem-resistant Pseudomonas in the text. The abstract (and conclusion) should be improve to clearly highlight the main findings and relevance.

1R. The sentence has been addeded “We considered blood, urine or respiratory specimens obtained with bronchoalveolar lavage (BAL) or bronchial aspirate (BASP), collected by all patients admitted in ICUs with or without COVID-19 infection”.

This sentence has been modified: “We found a higher frequency of infections due to methicillin resistant (MR)-Staphylococci, vancomycin-resistant Enterococcus faecium, carbapenem-resistant Acinetobacter baumannii and Candida parapsilosis in COVID-19 positive patients admitted in ICU compared to those COVID-19 negative. Carbapenem-resistant Pseudomonas aeruginosa was more frequently isolated from patients admitted in no-COVID ICUs”.

  1. In M&M, the paragraph on CA-UTI (line 108-line124) is much too detailed, and should be shortened using a reference while mentioning the main microbiological reading used. For e.g., Line 118, Are the signs and symptoms compatible with CA-UTI crucial for the present study (or relevant in the context of critically ill COVID-19 patients?). In M&M, authors should also clarify 1) how resistance to antibiotics were performed and 2) how the drug costs (economic analysis) were calculated.

2R. As suggested by Reviewer, we modified the text at lines 126-128 with this sentence “All criteria to confirm CA-UTI in several clinical conditions and with particular microbiological findings were considered according to international classifications and reports”. References 23,24 have been added.

1R) This sentence has been added: “Each isolate was identified by Matrix Assisted Laser Desorption Ionization Time-of-Flight (MALDI-TOF) by VITEK® MS (bioMerieux). Antibiotic susceptibility testing was performed by Card AST-N376 and N397 by VITEK® 2 instrument (bioMerieux). Carbapenemase-producer Enterobacteriales (CPE) strains were confirmed by microdilution (Sensititre™, Thermo Fisher Scientific) EURGNCOL and DKMGN plates. Phenotypical CPE resistance was confirmed by synergic test diffusion Diatabs™ (Rosco Diagnostica) on Muller-Hinton agar (Vacutest-Kima). Genotyping test for CPE resistance was performed by RT-PCR (GeneXpert®) with Xpert® Carba-R test (Cepheid Inc.). MIC values were interpreted according to current European Committee on Antimicrobial Susceptibility Testing (EUCAST) clinical breakpoints.”

2R) About drug costs calculation, the paragraph “2.2. Antimicrobial consumption and costs has been added with these sentencences: “The data of the antibiotic consumption were normalized to DDD/100 bed-days for inpatients. This information was calculated independently by the Pharmacy Service based on dispensing rather than administration data, using our own computer system that estimated the DDD [25 Rodríguez-Baño J., Paño-Pardo J.R., Alvarez-Rocha L., Asensio Á., Calbo E., Cercenado E., Cisneros J.M., Cobo J., Delgado O., Garnacho-Montero J., et al. Programas de optimización de uso de antimicrobianos (PROA) en hospitales españoles: Documento de consenso GEIH-SEIMC, SEFH y SEMPSPH. Enferm. Infecc. Microbiol. Clin. 2012;30].”.

Crude costs of antimicrobial drugs were collected from the administrative pharmacy data.

  1. The names of the three cohorts are very similar, and as such slightly confusing, i.e. COVID-19 ICU, no-COVID-19 ICU, and ICU-2019. The name also changes from COVID-19 ICU to ICU-COVID-19, and towards the end of the manuscript as SARS-CoV-2 ICU. Please harmonize throughout the text.

3R. Names of three cohorts were modified and harmonized according to the Reviewer’s suggestion.

  1. There is no information about the patients in the three cohorts, therefore it is difficult to estimate how comparable they are. Please provide more information on patients characteristics in M&M. Any information available related to the conditions/causes for the admission of “no-COVID-19 ICU” and “ICU-2019” patients. Admission for other respiratory infections? Were these patients also connected to respiratory support system or other support systems?

4R. The ICUs were a mixed medical-surgical ICU and a dedicated COVID-19 ICU at University Hospital of Ferrara, Italy. All their patients needed mechanical ventilation settings included constant-flow controlled ventilation. This sentence has been added in Materials and Methods paragraph: “The ICUs were a mixed medical-surgical ICU, ICU-noCOVID and ICU-2019, and a dedicated ICU for COVID-19 positive patients (ICU-COVID) at University Hospital of Ferrara, Italy. All patients admitted in these ICU wards needed mechanical ventilation settings included constant-flow controlled ventilation.”.

5.There is no information about data collection; from where does these data come from? Hospital?

5R. The data of the microbiological analyses performed at the Clinical Microbiology Laboratory of the University Hospital 'S. Anna 'di Ferrara collected in the database in the periods March-April 2019 for ICU-2019 and March-April 2020 for ICU-COVID and ICU-noCOVID.

  1. In the current form, the discussion does not provide added-value compare to the results section. It mostly paraphrase the results section. Only 2 references are used throughout the full discussion.
  2. Table 4 is not mentioned in the text. Please adapt.

7R. Table 4 has been mentioned in the text at line 267.

- The first sentence of the conclusion is rather surprising, as authors mentioned that respiratory co-infection cannot be excluded in COVID-19 patients, but their data show no significant difference between the three cohorts. The structure should be adapted, highlighting the main findings.

Specific comments:

Line 58: “The clinical manifestations of Coronavirus disease 2019, COVID-19)” => “The clinical manifestations of Coronavirus disease 2019 (COVID-19)”.

R: We modified the sentence as suggested by Reviewer: “The clinical manifestations of Coronavirus disease 2019 (COVID-19)…”

Line 18, Line 49, and throughout text: It is not clear why authors use both COVID-19 and COVID-19 nomenclature (when referring to the disease). Homogenized throughout text using WHO nomenclature for both disease and virus: https://www.who.int/emergencies/diseases/novel-coronavirus-2019/technical-guidance/naming-the-coronavirus-disease-(COVID-2019)-and-the-virus-that-causes-it

  1. R. The nomenclature has been adapted has suggested by Reviewer using WHO nomenclature.

Line 79: “patients. In" =>  "patients in".

R: point has been deleted.

Line 75: “Primary objectives were (i) the number of microbiological […]” => Please adapt this sentence containing the objectives. A word is likely missing.

  1. R. The sentence has been revised as “Our study examined microbial cultures performed in COVID-19 positive patients admitted to an ICU from February 22, 2020 to April 30, 2020.”.

Line 113: Define CA-ABU

R: CA-ABU has been deleted. We modified the sentence as suggested by Reviewer.

Line 115: "CA-asymptomatic bacteriuria" => "CA-asymptomatic bacteriuria (CA-ASB)"

R: CA-asymptomatic bacteriuria has been deleted. We modified the sentence as suggested by Reviewer.

Line 149: “in the 2019” => “in 2019”

R: the sentence has been modified harmonizing the names of three cohorts studied.

Line 160: Please clarify the following sentence: “No relevant data emerged regarding the number of positives for MRSA, ESBL-positive and carbapenem-resistant and carbapenemase-producing Enterobacteriaceae in the three cohorts." No colonies were isolated or no difference of CFU/mL between the cohorts?

  1. R. The sentence has been revised as “No relevant data emerged regarding the number of positive cultures for MRSA…”.

Line 164: Define DDD in text here (or prior in M&M).

Line 164R: DDD has been defined in M&M.

Line 166: The tables numbering is odd; Going from table 3 directly to table 6 in the text. Please adapt with incrementing numbers.

R: The tables numbers have been adapted with incrementing order.

Line 219:  […] may have to led to reduced compliance with […] => Clumsy English. Please adapt.

R: We adapted all sentence “Furthermore, the proper use of personal protective equipment can be challenging and can lead to lower compliance with aseptic techniques in the management of intravascular devices, also favoured by the recruitment of new emergency health personnel.”.

Line 234: In this list, order the drug in a logical manner (e.g. from most used to least used).

Line 234R: The list of drugs has been ordered from most used to least used.

Table 6: “Pazients” => “Patients”

Table 6R: We modified “Patients”.

Line 59 to 63: Please give references.

R: Reference has been added.

Reviewer 3 Report

The article entitled “Co-Infections in Critically Ill Patients with or without CoVID-2 19: a Comparison of Clinical Microbiological Culture Findings.” by Cultrera et. al, tried to draw a correlation between COVID-19 and secondary infection in critically ill ICU patient. The findings is interesting but have limited relevance to the field.

Manuscript is poorly written, many unstructured sentences with missing verb form and other basic grammatical errors makes the manuscript hard to understand.

In conclusion, bigger study group is required  to draw firm conclusion.

Line 68: the latter are not are proven

Line 69: cultures microbiological performed in CoVID-19

Sentence Line 75-76:

CoVID-19:to  COVID-19

microbiological culture: microbial culture is preferred term.

Line 147: performed show : performed shows

ICU-CoVID-19 emerges compared to

In Result section after table 3, author jumped to table 6 and table 4 &5 are not present neither mentioned. Table should be numbered according to the results.

Line 167: Costs for antimicrobials increased with particular reference to carbapenems, piperacillin / tazobactam, colistin, linezolid, tigecycline and third generation cephalosporins while there was no increase in ceftazidime / avibactam compared to the previous year.   Sentence is not clear, grammatical error

Author Response

Reviewer 3:

The article entitled “Co-Infections in Critically Ill Patients with or without COVID-2 19: a Comparison of Clinical Microbiological Culture Findings.” by Cultrera et. al, tried to draw a correlation between COVID-19 and secondary infection in critically ill ICU patient. The findings is interesting but have limited relevance to the field.

Manuscript is poorly written, many unstructured sentences with missing verb form and other basic grammatical errors makes the manuscript hard to understand.

In conclusion, bigger study group is required  to draw firm conclusion.

Line 68: the latter are not are proven

Line 69: cultures microbiological performed in COVID-19

Line 69R: The sentence has been modified as suggested by Reviewer in “Our study examined microbiological cultures performed in COVID-19…”.

Sentence Line 75-76:

COVID-19:to  COVID-19

R: we changed throughout the text COVID-19 with COVID-19.

microbiological culture: microbial culture is preferred term.

R: According to suggestions of the Reviewer, we modified “microbiological culture” with “microbial culture”. We made the change throughout the manuscript text.

Line 147: performed show : performed shows

Line 147R: The sentence is “The analyses performed show an increase…” not “The analysis performed…”.

ICU-COVID-19 emerges compared to

In Result section after table 3, author jumped to table 6 and table 4 &5 are not present neither mentioned. Table should be numbered according to the results.

  1. Tables have been ordered.

Line 167: Costs for antimicrobials increased with particular reference to carbapenems, piperacillin / tazobactam, colistin, linezolid, tigecycline and third generation cephalosporins while there was no increase in ceftazidime / avibactam compared to the previous year.   Sentence is not clear, grammatical error.

  1. R. Sentence has been changed as Reviewer suggested.

Reviewer 4 Report

The manuscript entitled “Mycological and molecular diagnosis and epidemiology of candidemia in hospitalized patients with coronavirus disease 2019 (COVID-19) in Tehran, Iran” aimed to investigate the number of microbiological cultures of blood, urine and bronchoaspirate / bronchoalveolar lavage samples. Moreover, to define the frequency of isolated bacterial and fungal species and the consumption of antimicrobials in ICUs in the two periods examined and in patients with or without CoVID-19. In addition, to evaluate the expenditure on antimicrobials used by each ICU in relation to the infections identified by culture tests.

Generally, the manuscript is well-written and included all aspects of the field. Data presented in the table are of relevance and information, however the manuscript requires revisions with gross English editing and confirmation or proven for my own questions.

More specifically

The major concern is that the impact of work is not presented enough in the manuscript, which is not focused on COVID-19 associated with fungal infections. Please clarify and use the relative references in line 62. For instances

  • Al-Hatmi AMS, Mohsin J, Al-Huraizi A, Khamis F. COVID-19 associated invasive candidiasis. J Infect. 2020 Aug 7:S0163-4453(20)30539-9. doi: 10.1016/j.jinf.2020.08.005.
  • Salehi M, et al. Opportunistic Fungal Infections in the Epidemic Area of COVID-19: A Clinical and Diagnostic Perspective from Iran. Mycopathologia. 2020 Aug;185(4):607-611. doi: 10.1007/s11046-020-00472-7.
  • Ahmadikia K, et al. The double-edged sword of systemic corticosteroid therapy in viral pneumonia: A case report and comparative review of influenza-associated mucormycosis versus COVID-19 associated mucormycosis. Mycoses. 2021 Feb 16. doi: 10.1111/myc.13256. Epub ahead of print. PMID: 33590551.
  •  

Page 5, line 10-11; authors mentioned “Few studies have addressed bacterial or fungal coinfections in COVID-19 patients” add references, however, so far numerous studies have been published. The following references missed

  • Lansbury L., Lim B., Baskaran V., Lim W.S. Co-infections in people with COVID-19: a systematic review and meta-analysis. J Infect. 2020;81(2):266–275. 

References regarding COVID19 in part of introduction are not really updated, replace add relative and informative references

  • Moradian N, et al. The urgent need for integrated science to fight COVID-19 pandemic and beyond. J Transl Med. 2020 May 19;18(1):205. doi: 10.1186/s12967-020-02364-2. PMID: 32430070

The author mentioned,” All patients had positive real-time PCR tests for COVID-19 and were suspected of co-infections based on them ???

Authors might consider the following question.

Please add significant data about which method was used for diagnosing each infections cases?

Diagnostic methods have a positive impact on clinical outcomes in patients with bacterial and fungal infections.

What are your inclusion and exclusion criteria for those patients? What criteria were used to include the cases with infection or colonization? Add reference(s).

Are those clinical isolates have been previously affected by antibacterial or antifungals?

Identification of Candida species has performed by????. Do you have availability to sequencing the isolates?

The discussion part should be taken into account by focusing on COVID19 associated fungal infections as well” Authors discussed more specifically about COVID19 in general, not COVID19 associated with fungal infection

What is your conclusion?

Authors should at least speculate what could be the importance of their finding in practical application.

What are the limitations?

Update references

Author Response

Reviewer 4:

The manuscript entitled “Mycological and molecular diagnosis and epidemiology of candidemia in hospitalized patients with coronavirus disease 2019 (COVID-19) in Tehran, Iran” aimed to investigate the number of microbiological cultures of blood, urine and bronchoaspirate / bronchoalveolar lavage samples. Moreover, to define the frequency of isolated bacterial and fungal species and the consumption of antimicrobials in ICUs in the two periods examined and in patients with or without COVID-19. In addition, to evaluate the expenditure on antimicrobials used by each ICU in relation to the infections identified by culture tests.

Generally, the manuscript is well-written and included all aspects of the field. Data presented in the table are of relevance and information, however the manuscript requires revisions with gross English editing and confirmation or proven for my own questions.

More specifically

The major concern is that the impact of work is not presented enough in the manuscript, which is not focused on COVID-19 associated with fungal infections. Please clarify and use the relative references in line 62. For instances.

  1. We modified the sentence and the relative references as suggested by Reviewer.
  • Al-Hatmi AMS, Mohsin J, Al-Huraizi A, Khamis F. COVID-19 associated invasive candidiasis. J Infect. 2020 Aug 7:S0163-4453(20)30539-9. doi: 10.1016/j.jinf.2020.08.005.
  • Salehi M, et al. Opportunistic Fungal Infections in the Epidemic Area of COVID-19: A Clinical and Diagnostic Perspective from Iran. Mycopathologia. 2020 Aug;185(4):607-611. doi: 10.1007/s11046-020-00472-7.
  • Ahmadikia K, et al. The double-edged sword of systemic corticosteroid therapy in viral pneumonia: A case report and comparative review of influenza-associated mucormycosis versus COVID-19 associated mucormycosis. Mycoses. 2021 Feb 16. doi: 10.1111/myc.13256. Epub ahead of print. PMID: 33590551.
  •  

Page 5, line 10-11; authors mentioned “Few studies have addressed bacterial or fungal coinfections in COVID-19 patients” add references, however, so far numerous studies have been published. The following references missed.

  1. We modified the sentence and the relative references as suggested by Reviewer.
  • Lansbury L., Lim B., Baskaran V., Lim W.S. Co-infections in people with COVID-19: a systematic review and meta-analysis. J Infect. 2020;81(2):266–275. 

References regarding COVID19 in part of introduction are not really updated, replace add relative and informative references

  • Moradian N, et al. The urgent need for integrated science to fight COVID-19 pandemic and beyond. J Transl Med. 2020 May 19;18(1):205. doi: 10.1186/s12967-020-02364-2. PMID: 32430070

The author mentioned,” All patients had positive real-time PCR tests for COVID-19 and were suspected of co-infections based on them ???

Authors might consider the following question.

Please add significant data about which method was used for diagnosing each infections cases?

Diagnostic methods have a positive impact on clinical outcomes in patients with bacterial and fungal infections.

  1. We added in MM diagnostic methods for bacterial and fungal diagnosis.

What are your inclusion and exclusion criteria for those patients? What criteria were used to include the cases with infection or colonization? Add reference(s).

  1. We considered all patients hospitalized in ICUs. We added reference about definition of infection vs colonization.

Are those clinical isolates have been previously affected by antibacterial or antifungals?

Identification of Candida species has performed by????. Do you have availability to sequencing the isolates?

  1. We added in MM diagnostic methods for fungal diagnosis. We have not availability to sequence the isolates.

The discussion part should be taken into account by focusing on COVID19 associated fungal infections as well” Authors discussed more specifically about COVID19 in general, not COVID19 associated with fungal infection.

  1. We modified as Reviewer suggested the “Discussion”.

What is your conclusion?

Authors should at least speculate what could be the importance of their finding in practical application.

What are the limitations?

Update references

  1. References have been added.

Round 2

Reviewer 2 Report

Minor revision:

Line 20: “with those had in a no-COVID-19”   =>   “with those in a no-COVID-19”

Line 65: “Some patients with COVID-19 due to severe respiratory complications need to be hospitalized and, in severe[…]”   =>   “Some patients with COVID-19 need to be hospitalized due to severe respiratory complications , and in severe[…]”

Line 80: “Primary objectives were (i) the number of microbial cultures of blood, urine and 80 bronchoaspirate / bronchoalveolar lavage samples, (ii) to define the frequency of isolated 81 bacterial and fungal species, (iii) the consumption of antimicrobials in ICUs in the two 82 periods examined and in patients with or without COVID-19”.   =>   “Primary objectives were to define (i) the number of microbial cultures of blood, urine and 80 bronchoaspirate / bronchoalveolar lavage samples, (ii) the frequency of isolated 81 bacterial and fungal species, (iii) the consumption of antimicrobials in ICUs in the two 82 periods examined and in patients with or without COVID-19.

Line 84: “Secondary objective was"    => "The secondary objective was”

Table 4 (Line 219): Please correct “Partie 28”, which I suppose is the number of patients and should be “Patients (n. 28)” ?

Author Response

Answers to Reviewer 2
Reviewer 2:

Minor revision:

Line 20: “with those had in a no-COVID-19”   =>   “with those in a no-COVID-19”

  1. The sentence has been changed as Reviewer suggested.

Line 65: “Some patients with COVID-19 due to severe respiratory complications need to be hospitalized and, in severe[…]”   =>   “Some patients with COVID-19 need to be hospitalized due to severe respiratory complications , and in severe[…]”

  1. The sentence has been changed as Reviewer suggested.

Line 80: “Primary objectives were (i) the number of microbial cultures of blood, urine and 80 bronchoaspirate / bronchoalveolar lavage samples, (ii) to define the frequency of isolated 81 bacterial and fungal species, (iii) the consumption of antimicrobials in ICUs in the two 82 periods examined and in patients with or without COVID-19”.   =>   “Primary objectives were to define (i) the number of microbial cultures of blood, urine and 80 bronchoaspirate / bronchoalveolar lavage samples, (ii) the frequency of isolated 81 bacterial and fungal species, (iii) the consumption of antimicrobials in ICUs in the two 82 periods examined and in patients with or without COVID-19..

  1. The sentence has been changed as Reviewer suggested.

Line 84: “Secondary objective was"    => "The secondary objective was”.

  1. As Reviewer suggested, we added “The secondary objective was…”.

Table 4 (Line 219): Please correct “Partie 28”, which I suppose is the number of patients and should be “Patients (n. 28)” ?

  1. I apologize for the writing mistake. We changed “Partie 28” with “Patients (n = 28). We changed also “n.” with “n =) in the table 4.

Reviewer 3 Report

The manuscript has improved significantly after the revision, few minor suggestions mentioned below could be incorporated before acceptance.

Line 74: Our study examined microbial cultures performed in COVID-19 positive patients admitted to an ICU from February 22, 2020 to April 30, 2020.

For table 6: The author could also mention in text or table itself the increase in cost could also be due to depreciation and price rise especially at the COVID time when the supply chain of drugs and other items were severely affected by lockdown.

Author Response

Reviewer 3:

The manuscript has improved significantly after the revision, few minor suggestions mentioned below could be incorporated before acceptance.

Line 74: Our study examined microbial cultures performed in COVID-19 positive patients admitted to an ICU from February 22, 2020 to April 30, 2020.

  1. The sentence has been revised as Reviewer suggested.

For table 6: The author could also mention in text or table itself the increase in cost could also be due to depreciation and price rise especially at the COVID time when the supply chain of drugs and other items were severely affected by lockdown.

  1. The sentence “The increase in costs was strictly related to the increased consumption of antimicrobials and not to the increase in direct costs or the reduction in drug supply affected by the lockdown.” has been added as Reviewer suggested.